# Child and Youth Health Literacy: A Conceptual Analysis and Proposed Target-Group-Centred Definition

**DOI:** 10.3390/ijerph16183417

**Published:** 2019-09-14

**Authors:** Janine Bröder, Orkan Okan, Torsten M. Bollweg, Dirk Bruland, Paulo Pinheiro, Ullrich Bauer

**Affiliations:** 1Centre for Prevention and Intervention in Childhood and Adolescence (CPI), Bielefeld University, 33615 Bielefeld, Germany; orkan.okan@uni-bielefeld.de (O.O.); torsten.bollweg@uni-bielefeld.de (T.M.B.); paulo.pinheiro@uni-bielefeld.de (P.P.); ullrich.bauer@uni-bielefeld.de (U.B.); 2Institute for Education and Care research in the health sector (InBVG), University of Applied Sciences Bielefeld, 33619 Bielefeld, Germany; dirk.bruland@fh-bielefeld.de

**Keywords:** health literacy, children, young people, adolescents, pupils, concept, definition, conceptual analysis

## Abstract

(1) Background: This article adopts an interdisciplinary perspective to analyse, examine, and reflect upon prominent health literacy (HL) understandings in childhood and youth. (2) Method: The conceptual analysis combined Rodgers’ and Jabareen’s approaches to conceptual analysis in eight phases. (3) Results: First, we present exploratory entry points for developing a child-specific HL understanding based on the six dimensions of a ‘health-literacy 6D model’. Second, we describe and reflect upon five meta-level dimensions covering the HL definitions and models for children and youth found in the conceptual analysis. Third, we integrate our findings into a target-group-centred HL definition for children and youth. (4) Discussion/Conclusion: This article raises awareness for the heterogeneity of the current conceptual HL debate. It offers a multidisciplinary approach for advancing the existing understanding of HL. Four recommendations for future actions are deduced from the following four principles, which are inherent to the proposed target-group-centred HL definition: (a) to characterize HL from an asset-based perspective, (b) to consider HL as socially embedded and distributed, (c) to recognize that HL develops both in phases and in flexible ways, and (d) to consider the multimodal nature of health-related information. Further research is necessary to test the feasibility and applicability of the proposed definition and conceptual understanding in both research and practice.

## 1. Introduction

From an early age onwards, children and young people are exposed to and have to deal with different, often complex health-related information and messages coming from various sources [1]. Health literacy, commonly defined as comprising a broad range of knowledge and competencies, can empower children and young people to seek, engage with, and use health information to enable and access health-promoting activities [2,3].

Through health literacy competencies, people become able to understand themselves, others and the world in a way that will enable them to make sound health decisions, and to work on and change the factors that constitute their own and others’ health chances [2].

Moreover, Fairbrother, Curtis, and Goyder [1] have drawn attention to children’s health literacy practices in accessing and understanding health information and their reliance on their embodied experiences. From a health promotion perspective, it is, therefore, essential to recognize children and young people as a target group of health literacy research and interventions: childhood and youth are regarded as foundational life phases for healthy human development and for personal health and well-being throughout adulthood e.g., [4]. Because crucial pathways for future health throughout the life course are already evolving and manifesting during childhood and youth [5], children should be supported by meeting their information needs and fostering their active involvement in their own health [6,7]. Accordingly, it is prominently voiced that failing to provide young populations with health literacy and health-promoting capacities would constitute an increased risk for the individual and society in terms of poorer health outcomes and higher costs [8,9,10].

Although child and youth health literacy has received increasing attention, the conceptual debate is characterized by a high degree of fragmentation and differentiation as well as the lack of a conclusive theoretical framework [11,12]. The conceptual work within this article was preceded by a systematic literature review of the available definitions and methods of health literacy for children and people younger than 18. This identified 12 definitions and 21 models that have been reported in detail elsewhere [11]. One main conclusion of this review was that there is limited consensus regarding any conceptual foundation and meaning of health literacy for children and young people, as well as a lack of clarity regarding target-group-specific health literacy characteristics.

Hence, in order to assess and adequately bolster health literacy during childhood and youth, it is important to be explicit and specific about what health literacy entails and means in these life phases. Conceptual understandings of health literacy should be evaluated critically and assessed in terms of their relevance, feasibility, and applicability for the target group [10,11]. In other words, transferring health literacy concepts developed for adults to children and young people requires proper consideration and evaluation of their applicability and how well they fit these life phases, as well as the target populations’ realities and needs [13,14].

Drawing on the results of the systematic literature review [11], the aim of the present research here was to examine and narrow down health literacy for school-aged children and young people in light of their age and developmental differences. First, this article applies a conceptual analysis designed to examine and reflect upon prominent understandings of health literacy in childhood and youth. Second, it explores dimensions for a target-group-centred conceptual understanding of health literacy in childhood and youth by drawing on relevant, interdisciplinary perspectives and then condensing these into a ‘differentiated’ multifaceted definition. Specifically, the article addresses the following research questions:What target-group-specific characteristics have to be considered in a differentiated, tailored understanding of health literacy in childhood and youth?What are the current challenges in the existing health literacy concepts for children and young people and what implications arise from these challenges?How can these considerations be transferred into a differentiated conceptual understanding?

## 2. Methods

This is an analytical and explorative article that uses a conceptual analysis to examine health literacy in childhood and youth. This analysis was conducted by combining Rodgers’ [15,16] evolutionary concept analysis with a more recent systematic ‘conceptual framework analysis’ approach from Jabareen [17]. According to Rodgers [15], ‘a concept is considered to be an abstraction that is expressed in some form, either discursive or non-discursive’. Concepts are subjected to continuous change and ‘through socialization and repeated public interaction, a concept becomes associated with a particular set of attributes that constitute the definition of the concept’ [15]. This is complemented by Jabareen’s [17] philosophical approach, which understands a concept as being defined by and consisting of its individual components that are not only distinct and heterogeneous but also related to one another and inseparable. Both approaches acknowledge that concepts overlap with other concepts. Jabareen’s [17] conceptual framework analysis is an iterative process, characterized by a continuous interplay between data collection, analysis, and reasoning. It requires a continuous comparison between different types of evidence, conceptual exploration, and discussion to constantly control the conceptual scope and level. Our conceptual/dimensional analysis was conducted along the phases reported in Table 1.

Phase 1 consisted of mapping the data sources identified and selected in two earlier research steps: First, in order to identify, explore, and analyse unique particularities and characteristics of children and young people that are of relevance for health literacy research and practice, we applied an iterative process to search for and analyse relevant, multidisciplinary literature from childhood studies, educational science, and sociology. We then structured the findings along six dimensions (the 6-D model) [13,18]. Second, we drew on the results of a systematic review of available conceptual understandings of health literacy for children and young people [11]. In Phase 2, the selected data were read and categorized, ensuring adequate representation of relevant multidisciplinary evidence from health science, sociology, childhood (development) studies, and literacy research. Next, in Phase 3, we analysed the identified body of literature to identify attributes and components of health literacy. The identified attributes and components were deconstructed and categorized in Phase 4. This entailed clarifying their characteristics, their assumptions, and their relation towards each other. In Phases 5 and 6, we integrated the results of the earlier phases by grouping relevant ones together and synthesizing them through an iterative process facilitated by reflective and analytical discussions within the research team. As an output, we identified a draft definition as well as the following five meta-level dimensions of current child and youth health literacy definitions and concepts: (a) content and attributes, (b) antecedents and contextual interrelatedness, (c) subject matter or topic purpose, (d) expected outcomes, and (e) life course relevance and target group sensitivity. The drafted target-group-centred health literacy definition and key conceptual dimensions were then discussed critically and validated communicatively at two expert workshops held in October 2016 and February 2018 (Phase 7). As part of this step, results were also presented and discussed within the academic setting though presentations at several national and international scientific conferences. The final Phase 8 entails continuing the critical review and reflection process and adopting or revising the concept according to need in light of new insights and evidence. Hence, this phase is still ongoing.

## 3. Results

### 3.1. What Special Characteristics of Children and Young People are Relevant for Health Literacy?

Childhood and youth are life phases in which needs, assets, and perspectives have unique characteristics [19]. Therefore, this section argues for the need to recognize children and youth as a distinctive target group for health literacy and to integrate their characteristics, needs, assets, and perspectives into target-group-specific health literacy definitions and concepts. Therefore, we reviewed relevant literature from multiple disciplines and structured the findings into a framework of six ‘Ds’—the so called ‘health-literacy 6-D model’. An earlier version of this model has been published elsewhere [13,18]. Drawing on and extending past ‘D’ models proposed by, among others, Rothman et al. [20], all dimensions start with the letter ‘D’ and represent exploratory entry points for developing a child-specific health literacy understanding (Figure 1). The entry points emphasize how and in what ways children and young people are a unique target group compared to the general adult population. Table 1 gives an overview of the core attributes of each dimension or ‘entry point’.

Each of the six dimensions of the 6-D model, described in Table 2, highlights relevant aspects that are important in order to elaborate on and explore health literacy holistically in terms of its relevance and meaning for children and young people. Essentially, we argue that, from a very young age onwards, children acquire experiences, form opinions, and develop their own unique understanding and meaning of not only the world around them, but also their health and well-being [1,21]. Nonetheless, intergenerational power relations are evident in every social interaction, and they influence children’s and young people’s roles regarding their health along with their active contribution to health-related decision-making processes. Hence, it is important to understand the nature and dynamics of these power structures for health literacy in a given social environment or situation and to create opportunities for children and young people to take ownership of and actively participate in their health literacy processes and actions.

### 3.2. Health Literacy of Children and Young People: Conceptual Analysis and Reflection

The following section summarizes key results from the conceptual analysis and the challenges it raises when trying to understand health literacy in the target populations. The following five meta-level dimensions were identified by drawing on available child and youth health literacy definitions and concepts [11]: (a) content and attributes of health literacy, (b) antecedents and contextual interrelatedness, (c) subject matter or topic purpose, (d) expected outcomes, and (e) life course relevance and target group sensitivity.

#### 3.2.1. Content and Attributes

Prevailingly, health literacy has been defined as a multi-dimensional, complex construct, entailing, among others, relevant skills and knowledge to seek and deal with health information and health-related decision-making in the health care, work, and other life settings [11]. Available definitions and models can be characterized along a continuum of two approaches that can be phrased as two questions, namely: (a) ‘what should a health literate person be able to do?’, hence the direct purpose or aim of health literacy; and (b) ‘what are the abilities, competencies and other attributes that characterize a health literate person or a health literate entity/community?’

Within the first approach, typical actions or tasks related to children’s and young people’s health literacy are:Receiving or actively seeking access to relevant information for one’s health through various personal or medial channels (e.g., after encountering a situation, problem, or demand that requires more information);Cognitively processing, concentrating (attention) on, and comprehending the information in order to understand its content;Critically appraising the credibility, accuracy, and relevance of information as well as interacting with that information by constructing meaning from the information and relating it to one’s situation or reality; andFollowing up on this information through health-related actions and decision-making [11].

The given action areas are broad and have varying degrees of complexity depending on the specific context or situation. Moreover, definitions and models representing this approach draw on the underlying assumption that health literacy skills, health information, and other components are per se relevant and meaningful for the target group, and that these can be used to achieve the defined health literacy actions and decisions [12].

Within the second approach, health literacy is described predominantly in terms of different combinations of individual cognitive abilities such as reading, writing, critical thinking, or information-processing skills. Whereas the focus on cognitive abilities prevails, health literacy has also been viewed as an umbrella concept encompassing, in addition to cognitive attributes, affective attributes (e.g., self-reflection, self-efficacy, motivation), operational or behavioural attributes (e.g., communicative and social skills), or specific technical skills (e.g., navigating the health care setting or system, technological information-searching skills) [11]. Moreover, different researchers have stressed that health literacy and its respective components represent ‘broader competence fields’ and not single skills e.g., [2,37] However, a variety of terms have been used to describe similar components without authors reporting their definition of such terms or the reasoning behind their choices. This is challenging, because labelling or defining health literacy as a set of skills creates a normative standard regarding what skills (levels) children and young people should possess or be compared to at a certain development stage in order to be considered health literate.

The heterogeneity within health literacy definitions and concepts regarding what health literacy is and which components it is composed of remains a crucial challenge for health literacy research, practice, and policymaking. Notably, health literacy has been criticized as being a ‘top-down’ or expert-driven concept [38]. Expert-driven approaches leave little room for individualized health literacy profiles, for recognizing children’s own assets, or for viewing them as active health agents and experts on their own lives from a young age onwards [13]. In a similar vein, Fairbrother et al. [1] stressed the need to move ‘beyond what children know’ towards research on ‘how children actively construct meaning from health information’ (p. 476). Table 3 provides a summary of the main findings and aspects raised. 

#### 3.2.2. Antecedents and Contextual Interrelatedness

There is a widespread emphasis on the importance of viewing health literacy in relation to not only a given environment but also the social context and situational demands [3]: health literacy is regarded as a product of situational requirements and individual attributes that are influenced by various social, demographic, and economic factors. Nonetheless, this widespread emphasis on the interrelatedness of context factors and health literacy is still considered and reflected only very implicitly in the most commonly used definitions for children and young people. In the systematic literature review [11], we summarized that contextual factors are still addressed most frequently as antecedent variables for health literacy. Hence, currently, little is known about how the contextual factors interact with one another, and how this interaction affects how children and young people can build up and use their personal health literacy skills. One approach to such interaction processes comes from Paek, Reber, and Larsccy [39] and Chuen, Miachel, and Teck [36], who have discussed the role of interpersonal and media socialization agents for adolescents’ health literacy from a sociological perspective. Contextual factors relevant for health literacy can be distinguished in:The interpersonal context such as the parental socio-economic status, parental education level, and the home setting;Situational determinants such as the degree of social support as well as influences from family and peers, the school and community setting, and the media; andThe distal social and cultural environment such as characteristics of the health and education system as well as political and social variables.

This is in line with results published by Malloy-Weir et al. [12] who pointed out that ‘there was variability, in terms of the context and/or time frames in which the various abilities and/or actions are believed to be important’ (p. 338). One example for the general population is the definition of health literacy by Kickbusch, Wait, and Maag [40] as the ability to ‘make sound health decisions in the context of everyday life at home, in the community, at the workplace, in the health care system, the marketplace and the political arena’. Moreover, Sørensen et al. [41] point towards being able to ‘make judgments and take decisions in everyday life concerning healthcare, disease prevention and health promotion’ (p. 13). Although both definitions state clearly the relevance of health literacy in multiple contexts, including a broad range extending from rather technical/medical settings to ‘everyday life’, both fail to integrate the complex interrelatedness between the individual and the structures in which that individual is embedded. Similarly, Malloy-Weir et al. [12] critically assessed that Kickbusch et al.’s [40] definition ‘does not take into account contextual factors that may limit/prevent “sound health decisions”.’ Table 4 provides a summary of the main findings and aspects raised. 

#### 3.2.3. Subject of Interaction: Health Information/Message

Health information, health messages, or health knowledge are placed at the core of most health literacy definitions and models for children and young people. However, very few of them specify what type, form, or mode of information and messages they are referring to [11]. Two dimensions surface with regards to health information: (a) how is ‘health information’ defined, and (b) what are its characteristics or nature with regards to how individuals interact with different modes and formats of information.

Regarding the terminology, Borzekowski [8], while not defining the term ‘health information’, provides case examples on topic-specific health information and highlights the multimodal nature of health information by stating that it can come from ‘a variety of sources’. Nutbeam [42] focuses on ‘information’ or ‘messages’ related to ‘different forms of communication’ including the communication content and method. Paakkari and Paakkari [2] refer to knowledge and competencies as a resource and ‘input’ as a central subject within health literacy.

Next, ‘health information’ as it is commonly used within health literacy concepts and definitions, incorporates the notion of being evidence-based, scientifically proven, or expert advice; hence, strongly implying some kind of ‘objectivity’. In other words, this evidence-based, fact-oriented perspective suggests that there is a ‘right’ way to interpret a piece of information and that this information is useful for further decision-making and actions. Not only is this perspective with its underlying assumption highly problematic, it also does not fit children’s and young persons’ everyday life realities: most children and young people report accessing health information through (a) their parents or other close adults who are typically lay persons, (b) the Internet, or (c) their peers [43,44]. Young people report using websites and video platforms such as YouTube as information sources for lifestyle and health-related topics such as food, sports, personal hygiene, and beauty [32]. Because much digital information is provided by private persons or companies with a direct or indirect commercial interest, it is crucial that the children and young people who access them are able to understand and critically analyse the information’s content, its potential purpose, and its intention. With regards to food-related health information, Fairbrother et al. [1] discovered that 9- to 10-year-old school children often access or receive diverse, sometimes contrasting, or even conflicting information through their parents, teachers and, less frequently, health professionals. Next, this underlying assumption on expert knowledge as being evidence-based ignores the complexity of many health topics and the tremendous importance of the source, format, and communication channel.

When zooming in on the characteristics of health information, a piece of information content may be packaged in different ‘modes’: textual, oral, aural, numeric, pictorial, or symbolic. Literacy and communication researchers have argued that information and messages are packaged in or composed of multiple, different modes. They call this multimodality and describe it as ‘the normal state of human communication’ [45]. In other words, health information is not neutral content; it is loaded with power, because there is always a communication or interaction taking place when encountering or dealing with a piece of health information.

When it comes to elementary school children, oral and visual modes of conveying information seem to resonate more than written text [1]. For complex digital environments, Grossen and Nürnberger found that children’s success in seeking information online depends strongly on whether it is presented in ways tailored to their target group, such as using many pictures [46]. Because prominent social media channels, including YouTube, Facebook, and Twitter all have multimodal designs, such as combinations of text-based, visual, and audio modes, young persons need to familiarize themselves with multimodal (health) information instead of learning how to use single modes such as information from print-based media [33].

These examples highlight the ambiguity in the terminology of ‘health information’, even though most health literacy definitions and models for children and young people place such terminology at the core of health literacy. Hence, what is meant or referred to by these terms within health literacy definitions and models remains largely unclear and unspecified. Table 5 provides a summary of the main findings and aspects raised.

#### 3.2.4. Purpose and Expected Outcome

Many models and definitions of children’s and young people’s health literacy follow a sequential design by illustrating an effect relationship between personal skills, health information, and the ability to use both for health decision-making and health itself [11,41]. Due to the strong emphasis on cognitive abilities—often highlighted as ‘functional’ health literacy skills e.g., [42] or information-processing skills—it is suggested that children and young people participate both rationally and actively in their health and health-related decision-making. Moreover, some regard health literacy as a social determinant of health itself, and have postulated that promoting health literacy could help to reduce health inequalities [3]. Up to now, however, there is still no consistent evidence base to support these assumed effect relationships. Moreover, these assumptions fail to give adequate consideration to the complexity and broader sets of factors affecting behaviour and behavioural change: the interdependencies between the subject and her or his social and life contexts and cultural factors, as well as affective and emotional aspects such as self-efficacy, self-determination, habits, and belief systems [11,12].

In addition, it is assumed in the literature that health information and health literacy abilities are per se relevant and useful for promoting one’s own health. This would imply that having access to information and abilities will lead to better decision-making. However, what is understood as ‘better decision-making’ and ‘actions for promoting one’s health’ is determined decisively by health experts and in terms of societal norms regarding ‘what is healthy’. Hence, such a line of argumentation falls in line with historical perspectives on functional literacy [47] in which the purpose of literacy is to be able to fulfil and succeed in the function or role an individual holds in society, such as accomplishing one’s work-related tasks and responsibilities as a citizen (e.g., voting, organizing one’s financial affairs). Within this historical perspective, functional literacy has been defined mainly by economic and employment contexts and the requirements these impose on the individual [47]. Such a perspective that emphasizes the role of the individual as a rational actor and aims for compliance with and fulfilment of prescribed societal demands conflicts strongly with other outcomes of health literacy for children and young people such as personal and social empowerment, participation, and equity. Moreover, this implication ignores the vast evidence surrounding the complexity of individual decision-making and behavioural change. Table 6 provides a summary of the main findings and aspects raised.

#### 3.2.5. Target-Group Characteristics

Children’s and young people’s characteristics and life stage considerations were lacking in half of the models and definitions assessed by Bröder et al. [11]. Moreover, when they were considered, these explorations and recognition of the characteristics remained on a very broad level, decisively incorporating an ‘external’, adult view on the target group’s situation and the relevance of health literacy for them.

Recent approaches that consider developmental factors in children’s and young peoples’ health literacy can generally be characterized as being based on successive stages that build upon each other e.g., [8,9]. The focus is especially on cognitive development aspects, namely ones focusing mainly on Piaget’s model of cognitive development [48] that distinguish between specific age-related stages of development in the skills and competencies children should be able to master and employ at a certain age. Borzekowski [8] and Sanders et al. [9] conceptualize health literacy within four skill areas (prose/document literacy, oral literacy, numeracy, and system-navigation skills), providing examples of activities for each development stage. Similarly, the US National Health Education Standards (NHES) [49] provide an extensive classification of the health literacy skills that students attending a certain school grade should achieve and that can be used to test them. Some researchers address target-group perspectives from a sociological perspective, proposing a health literacy socialization model [39] or highlighting contextual influences within a socio-ecological model of health literacy for adolescents [50].

Besides focusing on children’s or young people’s health literacy as an individual attribute, many studies have also addressed the health literacy of persons close to the child such as caregivers, mothers, parents, and teachers (see [51]). Because these persons are certainly important contributors to children’s or young people’s health and health literacy, researchers have proposed that child and adolescent health literacy should be regarded as the product of both individual health literacy skills and the skills or resources available in the proximal social context—namely, the adults, peers, or institutions that young people trust. Among others, this is referred to as ‘collective’ [9] or ‘distributed’ [52] health literacy. This would entail not looking at an individual’s competencies and skills but at (a) the sum of health literacy resources available within this proximal social context, and (b) how these resources are then applied and used by whom within such a social entity in a specific situation—given the present power dynamics, layers of autonomy, and decision-making habits and cultures (for further information, see [21,53]). Mostly, and above all, this includes encouraging children and young people to express their views and ideas and to participate actively. Moreover, children from a young age onwards are to some extent already involved in their own self-care. In practice, however, manifest power relationships, time constraints, or other barriers lead to children being excluded from active participation; for instance, Coyne et al. [54] observed that while many children in Irish clinical care settings expressed a strong motivation to participate in health decision-making, a large proportion of them felt that they actually were not included. Hence, from a children’s rights perspective, we find it crucial to promote rights to participate while taking into account their right for protection and care [18]. Table 7 provides a summary of the main findings and aspects raised.

### 3.3. Proposing and Discussing a Differentiated Understanding of Health Literacy

The results of the conceptual analysis and the exploration of target-group characteristics in the 6-D model can be used to conclude that there is a need for a differentiated concept of children’s and young people’s health literacy that is better tailored to their specific characteristics. Moreover, the challenges identified above also need to be addressed—or at least reflected—in the differentiated concept. This section describes an approach designed to advance the available health literacy understanding by proposing a target-group-centred health literacy definition for children and young people:
Health literacy of children and young people starts early in life and can be defined as a social and relational construct. It encompasses how health-related, multimodal information from various sources is accessed, understood, appraised, and communicated and used to inform decision-making in different situations in health (care) settings and contexts of everyday life, while taking into account social, cognitive, and legal dependence.As such, health literacy is observable in children’s and young people’s interaction and practices with health-related information, knowledge, messages in a given environment (so called ‘health literacy events or interactions’), while encountering and being promoted or hindered by social structures (in micro, meso, and macro contexts), power relationships, and societal demands.

This definition recognizes health literacy as a combination of complex processes related to how children and young people seek and interact with health-related information in different contexts in their daily lives. These processes are becoming increasingly important in a society that depends on digital communication. Moreover, it encompasses a public health and health promotion perspective on health literacy by explicitly stating the concept’s relevance in different situations in everyday life and health (care) settings. It integrates characteristics of the respective target groups, recognizing not only children and young people as social beings in their own right but also the need to achieve a balance between their participation needs and their protection needs.

The relatedness and contextual embeddedness of health literacy is placed at the core of this definition by recognizing individual and distributed resources within given structures. In addition, socio-ecological factors, especially those underlying social practices and the persistence of habitual mechanisms, as well as the entanglement with the milieu-specific conditions of social background are considered to be factors exerting a strong influence on health literacy in childhood and youth.

On the outcome and impact level, it is assumed that health literacy leads to being engaged with personal and social health (decision-making) and that it has a high capacity to determine the development of skills relating to the management of one’s own health. Therefore, health literacy may also be considered as a variable for influencing health over the life course.

The proposed target-group-centred health literacy definition integrates three conceptual dimensions that are interrelated but need to be operationalized independently:
1. **Individual Health Literacy Assets:** namely, the personal cognitive and habitual characteristics/attributes including the child’s independent knowledge along with abilities such as the ability to change, belief systems, cultural norms, and motivations.*2.** Social Health Literacy Assets:** namely, the social and cultural resources one can access* via *present social support structures in the close social environment (family/peer/community context). This also points to the importance of the health literacy available to individuals and groups within their social environment and that, as such, is also part of the children’s health literacy.*3. **Situational Attributes of an Occasion in which Health Literacy is Relevant:** namely, characteristics and demands of a given environment in which health literacy interactions—specifically the interaction with information or the health care setting—take place and that promote or hinder children in making use of individual and social health literacy assets. This, in turn, influences their agency and their real opportunities to practise and engage in health literacy interactions in their everyday lives.

## 4. Discussion

This article aimed to examine, reflect on, and clarify a conceptual understanding of health literacy in children and young people from various interdisciplinary perspectives by asking what are the distinctive components of a child-centred health literacy understanding. Based on the available literature, we have argued that within health literacy research, perspectives on childhood and youth are influenced decisively by adult or expert perspectives, while, at the same time, largely ignoring target-group-specific characteristics (as pointed out in the 6-D model). Hence, developing a differentiated understanding of children’s and young people’s health literacy requires us to move beyond perceiving them from an adult perspective that claims to act in their best interest. The results of the conceptual analysis were then used to develop a differentiated, target-group-centred definition. The proposed definition will now need to be discussed further, and its validation and applicability will need to be tested in different cultural contexts—within and across countries—as well as in distinct health-related settings. Nonetheless, four implications can be deduced from the present definition and conceptual analysis. These can serve as guiding principles for future health literacy research and practice:

### 4.1. Characterizing Health Literacy from an Asset-Based Perspective

Nutbeam [42] has pointed out that health literacy can be an asset for one’s personal health. However, an asset-based approach e.g., [55] to health literacy goes beyond this perspective. It entails recognizing any factor (or resource) that enables or enhances the health literacy of individuals or groups. Hence, the asset-based perspective shifts from a deficit focus looking at problems and needs towards a resource-oriented perspective on health and well-being. It does this by emphasizing the strengths and capabilities of an individual or a community [56]. Therefore, an asset-based perspective includes exploring and identifying a person’s or community’s practices for accessing, understanding, appraising and applying information relevant for their health instead of predefining the necessary skills. What personal strategies and resources do children and young people possess for dealing with health information and messages? What contextual conditions are necessary to promote their abilities to do so, and what strategies are known to promote these assets? What assets related to health literacy are located within the individual, the immediate, direct surroundings/community, the broader community, or the distant context? These are crucial questions when applying an asset-based health literacy approach.

### 4.2. Considering Health Literacy as Being Socially Embedded and Distributed on Individual, Family, and Social Levels

Although health literacy has been regarded as a product of inherent personal skills and contextual demands [3], we have illustrated the need for a nuanced recognition of its complex contextual interdependencies. This entails considering a three-fold relational concept: (a) individual attributes, (b) social or contextual attributes, and (c) the situational attributes and characteristics when health literacy practices occur. Therefore, we regard health literacy not as a product (what?) but as a process and social practice (how?) of engaging with one’s health by seeking, dealing with, and using information for health-related decision-making. Moreover, we consider health literacy as being socially embedded, because the situational attributes and the demands imposed upon the individual by given cultural and social structures strongly influence that individual’s real opportunities to practise health literacy in a specific situation. These opportunities depend decisively on whether health literacy assets are available, accessible, and relevant to the individual when needed. Children’s and young people’s opportunities for being health literate are strongly influenced by the characteristics of intergenerational relationships, as well as the social roles attributed to them by adults, peers, and society within everyday interactions such as those between teachers and students or between doctors and child patients. Within such intergenerational negotiations, children and young people’s social status is consolidated or challenged. Being aware of these, sometimes implicit, intergenerational processes is key to understanding how children and young people are perceived, and which difficulties and opportunities they face when practising health literacy and actively promoting their health and well-being [14]. A nuanced milieu perspective on the children’s family milieu and their social and cultural capital can help to understand how personal health literacy is acquired and practised within the given social structures and mechanisms [13]. With regard to immigrant adolescents’ health literacy, Santos et al. [57] have proposed a similar focus, namely to view health literacy as a socially situated practice, using a bike-riding metaphor:
Bike-riding requires the coordination of many working parts (gears, brakes, handle bars, and pedals). Each part has a unique purpose, but their contributions have little meaning apart from the whole. Similarly, we view an immigrant adolescent’s process of becoming ‘health literate’ as an evolving coordination of many working parts (e.g., reading skills, math skills, form-filling skills, linguistic choices, digital tools, or interactions that involve any of these skills and tools). The significance of these parts cannot be accurately understood when apart from the social, cultural, and historical context in which immigrant children are growing up.([57] page 4)

### 4.3. Recognizing that Health Literacy Starts Early in Life and Develops in Flexible Ways

Many researchers have stressed the concept’s relevance for all ages and life phases, and they link health literacy to lifelong learning e.g., [41]. This is especially valid, because the amount of available information increases significantly with age and the complex (digital) information landscape is changing rapidly. It requires a certain agility in individuals and communities to adapt to changing demands. Whereas some personal abilities, experiences, and community capacities may be transferable and adaptive to new situations and circumstances, some are highly contextual and culture-dependent, making them difficult to transfer. Some may even be lost over time. This requires a person to adopt and learn new knowledge, resources, and skills throughout the life course. Therefore, we propose considering health literacy as a multifaceted socio-cultural learning process (and not a purely cognitive one), because it is built up through personal experiences and interactions within a given environment. Whereas developmental stages may offer a reference point, they should not be seen as nominal categories for classification and evaluation. In other words, predefining skill categories for different age stages may offer a guiding orientation for what health literacy skills a child may possess at a certain age, but this approach needs to be complemented with a differentiated, child-centred picture [14]. Children and young people are beings in their own right with individualized developmental histories and portfolios. Next, when deriving health literacy profiles and levels, these should be adaptable to constantly changing personalities and reflect increasing maturity and autonomy processes as well as changing cognitive and social skill levels and social-ecological conditions [14].

### 4.4. Considering that Health-Related Information is Multimodal, Complex, and Power-Loaded

By using the term ‘health-related’ information, we aim to stress that alongside evidence-based sources, information is received and retrieved from various, often informal, sources, including lay persons whom one trusts or even one’s own embodied experiences [43]. On the most basic level, the latter include a person’s own embodied messages about themselves, their bodies, and their minds as well as their place and role in a health context or situation. Hence, information may be highly subjective, controversial, and complex. When encountering a piece of information, meaning-making processes confirm, integrate, or reject it depending on whether it conflicts with or confirms a person’s values, attitudes, and belief systems [1]. In addition, it is crucial to recognize the multimodality of health-related information or messages, that information is power-leaded, and that there is always some sort of communication or interaction with this information [45]. Moreover, there are textual, oral, aural, numeric, pictorial, or symbolic modes of packaging a piece of content. Digital or media information, in particular, is composed of several modes (multimodal) such as visual, aural, or linguistic elements [45]. Because literacy practices are multimodal by their very nature, this requires the individual or group to be able to compensate for a lack of certain skills by drawing on other personal or social assets in innovative and creative ways and thus even to overcome limiting structural conditions [47]. At the same time, these are examples of culture- and context-dependent skills: a person who is able to read and write in one language is frequently not able to read and write in another language. Therefore, there is a need to distinguish between culturally independent and culturally dependent modes of interacting and communicating health-related information.

## 5. Conclusions

The primary aim of this article has been to raise awareness for not only the heterogeneity of the current conceptual debate on health literacy, but also the gaps and shortfalls in adequately representing target-group characteristics. The proposed child-centred health literacy understanding takes a sociological view on childhood and youth. This approach emphasizes how children and young people access health-related information, derive meaning from this information, and engage in health-related decision-making processes while being embedded within complex social contexts. Whether the proposed definition and conceptual understanding prove feasible to guide future health literacy research as well as actions to promote health literacy in childhood and youth remains an open question and calls for further research. This will include the analytical testing, operationalization, and practical evaluation of the definition and the guiding principles described here.

## Figures and Tables

**Figure 1 ijerph-16-03417-f001:**
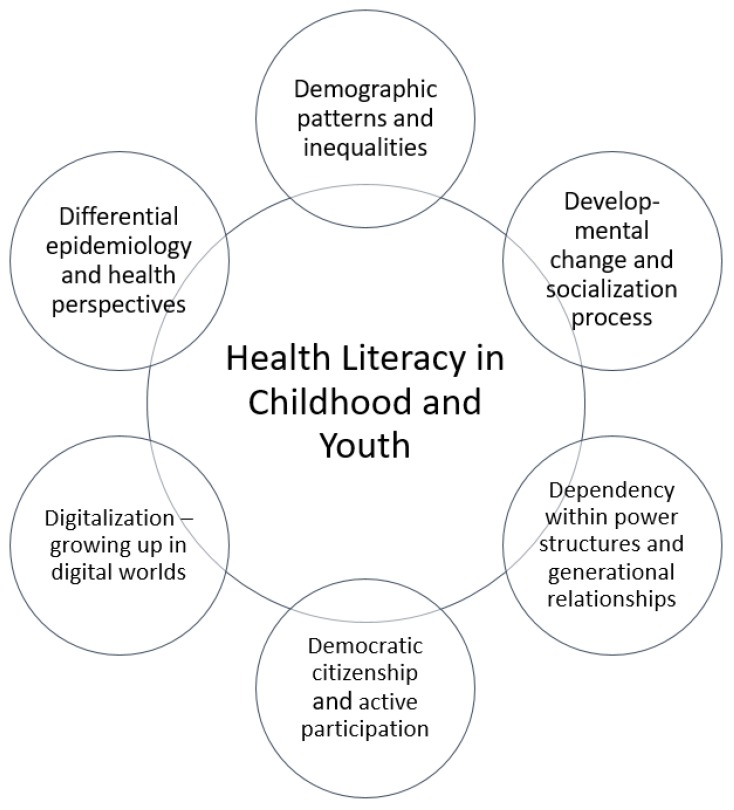
Health literacy 6-D model, highlighting six target-group-oriented health literacy dimensions.

**Table 1 ijerph-16-03417-t001:** Steps in the conceptual analysis (revised approach drawing on Rodgers [11] and Jabareen [13]).

**Phase 1**	Identifying and choosing the concept for analysis and mapping the selected data sources
**Phase 2**	Extensively reading and categorizing the selected data
**Phase 3**	Identifying and naming the dimensions and components of a concept
**Phase 4**	Deconstructing and categorizing the concept’s attributes, characteristics, and assumptions
**Phase 5**	Integrating the components
**Phase 6**	Grouping, synthesizing, and resynthesizing the dimensions
**Phase 7**	Validating the results
**Phase 8**	Identifying hypotheses and implications for future research and development

**Table 2 ijerph-16-03417-t002:** Description of dimensions of the health literacy 6-D model (for an earlier version, see [13,18]).

‘6 D’s’	Description
**Differential epidemiology and health perspectives**	‘Health’ or ‘well-being’ and ‘disease’ or ‘illness’ are culturally loaded concepts that are open to being interpreted and constructed socially. Hence, their meaning may differ within and between individuals, age groups, cultures, and professions (the ‘expert’ and the ‘lay person’) [22]. Children and young people understand and attribute meaning to the concept of health, of being healthy, or of being well by drawing on their personal embodied experience and their interpretation or uptake of articulated health-related beliefs and attitudes in their proximal social surroundings [1]. Although children and young people partly suffer from similar diseases and are exposed to similar health risk profiles as adults, some disease and health risks are highly age- or development-specific and are related to an increased vulnerability for exposure [4].
**Demographic patterns and inequalities**	Children and young people are especially vulnerable to social and health inequalities, because their health is influenced by a multitude of complex and interrelated factors in their proximal and distant social environment [4,23]. They are the age group with the highest poverty risk according to socio-demographic characteristics [24]. Factors such as low family socio-economic status, poor living conditions, poor access to higher education and social support structures, as well as having a migration background are associated with an increased risk of educational disadvantage; lack of skills, knowledge, and competencies; or even psychosocial developmental disabilities [4]. Children and young people exposed to these factors face a two times higher risk of obesity and rate their subjectively perceived health status and quality of life below the age group’s average [24,25].
**Developmental change and socialization process**	Childhood and youth are life phases in which essential biological, cognitive, psychological, emotional, and social development processes take place [26,27]. Every developmental phase is accompanied by specific developmental features, typical challenges, and social expectations e.g., [28]. Children and young people have to handle these expectations in order to shape their development process beneficially, and this advances their maturity and autonomy [26,27]. Apart from cognitive development aspects, namely the skills and competencies children should be capable of mastering and employing in the context of health literacy at a certain age or developmental stage, it is crucial to also recognize the sociological and psychosocial development processes that are taking place [14].
**Dependency within power structures and inter-generational relationships**	Whereas children and young people rely, to the extent they are dependent, on their parents’ assistance, competence, economic resources, and social support, they, at the same time, actively engage in and form their own social world/realities [21,29]. Intergenerational power relations and conflicts are evident when children interact with adult society, and this reflects the unequal distribution of power between children and adults [30]. Characteristics of generational order and social position are negotiated on a daily basis within peer groups as well as between children, adolescents, and adults [21].
**Democratic citizenship and active participation**	Children and young people have a right to be informed, to participate actively in their own health (decision-making), to access health information, and to have this information presented to them in understandable and appropriate manners [1,13].They are embodied beings and social actors within their own right who encounter and engage in health information and health-relevant situations on a daily basis [6].Children and young people’s agency can be characterized as contingent on the responsiveness of and the opportunities available within the networks of actors and the structure of the social [31].
**Digitization/Digital worlds of growing up**	Many children and young people grow up in highly digitized and media-saturated settings [32,33]. Some refer to children and young people as ‘digital natives’, implying that children ‘naturally’ learn and become socialized with digital media formats because digital media are an integral component of their daily lives [34]. Because they encounter and access health information in various or even multiple digital forms and formats, considering the opportunities and challenges in digital and media settings with their various multimodal formats is crucial for understanding children’s and young people’s health literacy and their health information seeking [35,36].

**Table 3 ijerph-16-03417-t003:** Summary of conceptual reflections regarding health literacy content and attributes.

Description of Findings	Prominent Argumentation Lines Identified in the Conceptual Analysis	Implications and Challenges Arising from the Findings
Two major perspectives on health literacy during childhood and youth:		Focuses on individual attributes and competence areas Implies that cognitive abilities lead to actions and personal agency Emphasizes cognitive attributes more than conative/personal/emotional ones Defines action areas very broadly, thus enabling individualization but creating challenges with regards to specifying and distinguishing health literacy from other concepts Is heterogeneous and defines normative standards
(a) Action- and output-focused perspectives	Action-oriented or output-focused health literacy approach: defining individual actions related to accessing, understanding, judging, and using health information	
(b) Skills-focused perspectives	Skills- or input-focused approach to health literacy that entails different combinations/sets of skills or broad competence areas	

**Table 4 ijerph-16-03417-t004:** Summary of conceptual reflections regarding health literacy antecedents and their contextual interrelatedness.

Description of Findings	Prominent Argumentation Lines Identified in the Conceptual Analysis	Implications and Challenges Arising from the Findings
Consideration of contextual influences and the relational character of health literacy for children and young people remains shallow	Contextual factors are antecedent or mediating factors for the health literacy of children and young people Individual abilities need to correspond with situational demands	Interaction or interrelated character of contextual factors and individual abilities is oversimplified Contextual interdependencies between setting-related lifestyles, habits, or dispositions remain insufficiently understood Further research is needed to understand how individual abilities can be and actually are applied in a situation requiring health literacy

**Table 5 ijerph-16-03417-t005:** Summary of conceptual reflections regarding ‘health information’ within health literacy.

Description of Findings	Prominent Argumentation Lines Identified in the Conceptual Analysis	Implications and Challenges Arising from the Findings
Health literacy is prominently defined as being centred around information or messages	‘Health information’ is the subject requiring health literacy abilities No distinction is made between ‘factual and tacit knowledge’ categories	Insufficient recognition of multimodal nature of information and communication channels Lack of specification of what health information is leads to ambiguity and reductions in meaning when it is operationalized

**Table 6 ijerph-16-03417-t006:** Summary of conceptual reflections regarding the purpose and expected outcomes of health literacy.

Description of Findings	Prominent Argumentation Lines Identified in the Conceptual Analysis	Implications and Challenges Arising from the Findings
Sequential effect relationship is proposed	Insufficient evidence base to support assumed effect relationships for the target group. Health information is generally useful for children and young people and their health Individual cognitive skills will lead to actions and health-promoting behaviours. Outcomes implying personal decisions and behaviours to comply with given social norms and standards conflict with health literacy notions aiming towards personal empowerment and participation	Evidence needed on the purported effect relationships Need to clarify the validity of the available evidence supporting the specific conceptual understanding (e.g., ‘functional’ or more complex, public health perspectives of health literacy)

**Table 7 ijerph-16-03417-t007:** Summary of conceptual reflections on health literacy and target-group characteristics.

Description of Findings	Prominent Argumentation Lines Identified in the Conceptual Analysis	Implications and Challenges Arising from the Findings
Target-group characteristics in available concepts considered mainly in terms of cognitive development theories and deficit-oriented approaches	Age-related stages or standards are defined in terms of the personal health literacy abilities a child or young person should have at a certain age or developmental stage The focus on cognitive developmental processes limits health literacy to cognitive aspects It is necessary to recognize the ‘collective’ health literacy skills that are distributed between the individual and her or his close environment	Shift towards recognizing health literacy as a complex and highly differentiated social learning process Recognition and integration of sociological development perspective and the six ‘D’ dimensions described above

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
