# Peer review of "Child and Youth Health Literacy: A Conceptual Analysis and Proposed Target-Group-Centred Definition"

_ijerph, 2019, doi:10.3390/ijerph16183417_

Round 1

Reviewer 1 Report

In the prsent study's pourpose was to analyse, examine and reflect upon prominent health literacy (HL) understandings during childhood and youth through interdisciplinary perspectives and to propose a target group-tailored HL understanding for children and young people. Authors suggested that there are four principles inherent to the proposed target group-centred HL definition that can be deduced into four recommendations for future actions: (a) to characterize HL from an asset-based perspective, (b) to consider HL as socially embedded and distributed at individual, family and social level, (c) to recognize that HL starts early in life and develops both in phases and agilely/flexible, and (d) to consider the complexity of health information by being multi-modal health-related information from various sources.

This manuscript was written well, however, it need to be consided several things.

(1) Abstract; What is this article's back grounds? I did not understand this in the abstract.

(2) There ware several previsous studies in this Issue, could you consider add the other reference in this article? i.e.

Ogi H et al., Healthcare (Basel). 2018 Apr 2;6(2). pii: E32. doi: Associations between Parents' Health Literacy and Sleeping Hours in Children: A Cross-Sectional Study.

Author Response

Thank you for your helpful feedback and recommendations. We revised the background section of the abstract. Thank you for the reference suggestion. As the cross-sectional study addresses the impact of parental HL for children’s sleeping hours, and hence not the HL of children directly, we did not reference it in the manuscript. If you have other recommendations regarding articles addressing children’s and youth’s HL directly – we look forward to them.

Reviewer 2 Report

Minor essential revisions are needed in the Abstract, Introduction part,Methods and the Results. Discussion and Conclusions are well balanced and supported by the data.

There are a few typographical and grammatical errors throughout the text, that need to be tidied up.  See lines 13 and 45, for example.

Needs some English corrections before being published.

The authors need to define the children and young people as well as childhood and youth as a specific life period. What are the age differences between these categories?

The method is appropriate but need to be  better described with sufficient details provided to replicate the work. Authors have to include  explanation of all eight phases mentioned in table 1.

The Results section is well organized using sub-headings in accordance to the study aims, so please omit the first sentence from the Results section. Line 118.

Lines 34-35.  Please, add other references for health literacy.

Lines 45-46. Here health-promoting attitudes, beliefs and behaviors are mentioned without specify what are they.  Please, add a definition.

Lines: 262,297. Reference 40 is not cited correctly in the text. It should be placed before 41, not after.

Lines 334-335. Here, please add the reference for Piaget’s model of cognitive development.

Author Response

Thank you for your helpful feedback and recommendations. We appreciate your note on the article’s complexity and that it requires some advanced insights into health literacy. We after an extensive discussion on whether to follow your recommendation of including a footnote on this issue, we decided against it. We find that the issue is pointed out in the introduction and the articles’ purpose. Moreover, we reference the relevant literature ‘essential literature’ (a.o. Malloy-Weir, Sorensen, Perry, Kickbusch et al. - the WHO solid fact publication) and also clearly hint to earlier publications this article draws upon. Hence, we would like to leave it up to the readers to decide whether they find it a helpful conceptual analysis and discussion or whether they might want to consult other literature.

We added an example related to the proximal social context and power-dynamics in the care setting on page 11. Yes, we highly concur with your observation. However, as this is analysis is taking on a rather abstract perspective, not clinging to one specific setting, we would like to see if and who the definition is being tested in the different (cultural settings). As there are many structural issues, resulting in sever health inequalities, in many countries, we would argue that health literacy practices may take different forms and sizes as there are other contextual and structural constrains but also maybe other assets available to use. We added a note regarding this issue in the discussion, stating that the proposed definition needs further validation and testing in different cultural settings. Thank you for your suggestion. While the 6-D Model highlights what could be, do Tables 3-7 focus displaying how the conceptual discussion is currently led. We added the reference to the 6Ds in the beginning of part 3. We added bullet points in tables 3-7 to increase their readability and checked the construction of the sentences. Thank you for the literature suggestions. We found the perspective by Santos et al. very helpful and added it to support the third recommendation on page 15.

Reviewer 3 Report

A strong contribution to the health literacy literature, providing an expanded conceptual framework on which future studies (i.e. interventions in school settings, as well as in 'third spaces' where adolescents work, study, and play).  

It seems clear that the intended readership of this article will be people who are already sufficiently familiar with the health literacy field/literature base.  The literature covers a great range of knowledge related definitions and conceptualizations of health literacy, without having to unpack terms or provide explanatory examples (e.g., what is meant by collective health literacy).  Seems important that the article point the novice reader (e.g., grad students who are new to the field) to essential background reading. Please consider adding an endnote to this effect, or cite specific useful background articles.

The next few recommendations aim to improve the accessibility of the article's content:

1.  Please consider including more examples that illustrate the relevance of health literacy practices,and/or the evolving health literacy competence of adolescents.  For example, lines 348-351 reference the "proximal social context, namely adults, peers or institutions young people trust in" as an important context in which to understand adolescent/child health literacy.  Excellent point, which very much echoes social views on literacy (reflected in the K. Perry citation). This would be a good place to include an example which illustrates the importance of attending to in the "proximal social context" -- power dynamics? decision-making patterns? the ways adolescents resist adult involvement? Similarly, an illustration or example would be useful around lines 320-322 which refers to children's/adolescents' "personal and social empowerment, participation and equity" in health care.  A lot of major and meaning-laden terms here that the reader would be more likely to appreciate if there were examples to show what these terms mean for adolescent decision-making in their own health care...

2.  The article references literature from a variety of contents (including but not limited to the U.S. context). The article could add a qualifying note, somewhere in the beginning, about whether and how the conceptual review work accounted for differences across countries, i.e., differences in the distribution of health status across socioeconomic groups, institutional realities (e.g., no universal health care coverage in the U.S.), and ideological priorities that shape priorities and solutions across countries. If the focus here is on thinking about adolescent health with respect to the global health agenda on health disparities, that would be useful to state, acknowledging that there is significant variation in the perceptions of health disparities affecting child health around the world.

3.  What is the connection between the domains (6Ds) in Table 2 and the analyses associated with Tables 3-7? It seems there are several links that can be made, such as the way we should be thinking differently about the agency of adolescents in health care.  This realization suggestions that the content in Table 2 should influence the organization of information in Tables 3-7.  Can more be said about the content in these tables and their connection to Table 2?

Also, Tables 3-7 contains lots of useful summary information but are rather hard to read -- bullet the different points? use parallel structures across points within a section (e.g., lead with verb or verb forms, rather than mixing verbs and nouns).  Avoid passive construction in the 1st column? In any case, the content seems understandable based on the prose alone, with the tables inviting confusion.

4.  Consider adding two references to the paper:

Manganello, J. A. (2008). Health literacy and adolescents: a framework and agenda for future research. Health Education Research, 23(5), 840–847. https://doi.org/10.1093/her/cym069   Santos, M., Gorukanti, A., Jurkunas, L., Handley, M., Santos, M. G., Gorukanti, A. L., … Handley, M. A. (2018). The Health Literacy of U.S. Immigrant Adolescents: A Neglected Research Priority in a Changing World. International Journal of Environmental Research and Public Health, 15(10), 2108. https://doi.org/10.3390/ijerph15102108    

Author Response

(The authors gave the same response as above.)
